∂ | **Open Peer Review** | Systems Biology | Research Article

# Metabolic model predictions enable targeted microbiome manipulation through precision prebiotics

Georgios Marinos,[1] Inga K. Hamerich,[2] Reena Debray,[3] Nancy Obeng,[2] Carola Petersen,[2] Jan Taubenheim,[1] Johannes Zimmermann,[1,4] Dana Blackburn,[5] Buck S. Samuel,[5] Katja Dierking,[2] Andre Franke,[6] Matthias Laudes,[7] Silvio Waschina,[8] Hinrich Schulenburg,[2,4] Christoph Kaleta[1]

**ABSTRACT** While numerous health-beneficial interactions between host and microbiota have been identified, there is still a lack of targeted approaches for modulating these interactions. Thus, we here identify precision prebiotics that specifically modulate the abundance of a microbiome member species of interest. In the first step, we show that defining precision prebiotics by compounds that are only taken up by the target species but no other species in a community is usually not possible due to overlapping metabolic niches. Subsequently, we use metabolic modeling to identify precision prebiotics for a two-member *Caenorhabditis elegans* microbiome community comprising the immune-protective target species *Pseudomonas lurida* MYb11 and the persistent colonizer *Ochrobactrum vermis* MYb71. We experimentally confirm four of the predicted precision prebiotics, L-serine, L-threonine, D-mannitol, and γ-aminobutyric acid, to specifically increase the abundance of MYb11. L-serine was further assessed *in vivo*, leading to an increase in MYb11 abundance also in the worm host. Overall, our findings demonstrate that metabolic modeling is an effective tool for the design of precision prebiotics as an important cornerstone for future microbiome-targeted therapies.

**IMPORTANCE** While various mechanisms through which the microbiome influences disease processes in the host have been identified, there are still only few approaches that allow for targeted manipulation of microbiome composition as a first step toward microbiome-based therapies. Here, we propose the concept of precision prebiotics that allow to boost the abundance of already resident health-beneficial microbial species in a microbiome. We present a constraint-based modeling pipeline to predict precision prebiotics for a minimal microbial community in the worm *Caenorhabditis elegans* comprising the host-beneficial *Pseudomonas lurida* MYb11 and the persistent colonizer *Ochrobactrum vermis* MYb71 with the aim to boost the growth of MYb11. Experimentally testing four of the predicted precision prebiotics, we confirm that they are specifically able to increase the abundance of MYb11 *in vitro* and *in vivo*. These results demonstrate that constraint-based modeling could be an important tool for the development of targeted microbiome-based therapies against human diseases.

**KEYWORDS** nutritional supplements, flux balance analysis, genome-scale metabolic models, *Caenorhabditis elegans*, *Pseudomonas lurida* MYb11, *Ochrobactrum vermis* MYb71, serine

Address correspondence to Christoph Kaleta, c.kaleta@iem.uni-kiel.de, or Hinrich Schulenburg, hschulenburg@zoologie.uni-kiel.de.

Georgios Marinos and Inga K. Hamerich contributed equally to this article. Author order was determined in order of increasing seniority.

Hinrich Schulenburg and Christoph Kaleta are joint senior authors.

The authors declare no conflict of interest.

See the funding table on p. 18.

While traditionally considered commensals, it is becoming increasingly clear that the microbial communities living within and on higher host organisms (i.e., the microbiome) are integral parts of host physiology. These microbial communities fulfill essential roles in pathogen protection (1), immune system education (2), production of vitamins (3), metabolism of xenobiotics (4, 5), and increase the plasticity of their host to

respond to evolutionary challenges (6, 7). Thus, the composition of the microbiota plays an important role in the well-being of hosts and its ability to respond to environmental challenges (8, 9), as indicated by the frequent shift of microbiota composition observed in the context of diseases (10) and exposure to environmental stressors (11). These observations and work in animal models have led to the development of the notion that the composition of the microbiome itself might be an important modulator of host health (12) as well as fitness and that a targeted modulation of microbiome composition might offer a completely new avenue for modulating the host.

It has been shown that dietary patterns (e.g., western diet) can affect the structure of the microbiome, however their effect is transient (13). Besides an influence of dietary uptake on microbiome composition, various approaches to modify the microbial communities of organisms have been used (14): the transfer of entire microbial communities from a donor to a recipient host (fecal transfer), the provision of compounds used by particular microbial species (prebiotics), the provision of microbial strains of natural or synthetic origin (probiotics), the provision of compounds of microbial origin (postbiotics), and combinations thereof (synbiotics). While fecal transfer is the most commonly used approach to modulate microbial community composition in animal models, it involves the transfer of entire microbial communities and therefore not only desirable traits. In consequence, there is only a single medical indication in which fecal transfers are routinely used thus far (15) and there are potential concerns about the safety and stability of the engraftment of transplants (16).

Although the concept of supplementation with pre-, pro-, or synbiotics has been commonly proposed as a method of microbial manipulation, these approaches are still severely limited. In humans, oligosaccharides are proposed as prebiotics to alter the microbial community on both the structural and functional level, since they can be fermented by microbial species, which in turn can proliferate and produce bioactive compounds such as short-chain fatty acids (17). However, there is only little knowledge about other classes of compounds that might alter the microbial composition. Another important aspect to consider is the specificity with which individual target microorganisms can be manipulated using prebiotics. Thus, while oligosaccharides are considered beneficial, there is often uncertainty about which species can degrade them, as multiple, unrelated species can degrade similar types of oligosaccharides (e.g., *Bacteroidetes*, *Lactobacilli*, and *Bifidobacterium* species can degrade fructans) (17, 18). In case of supplementation with probiotics and synbiotics, other concerns have been raised. For instance, the ability of specific Bifidobacteria to stably colonize the human gut was only reported for ⅓ of the participants in a clinical trial (19). Moreover, introducing exogenous bacteria can have unwanted side effects. For example, some of the *Bacillus* species that were used as probiotics in aquacultures were found to carry virulence and antibiotic-resistance genes (20). Hence, to overcome these limitations in current microbiome manipulation approaches, we need to focus on prebiotics with well-controlled modes of action (e.g., based on known microbial pathways that can degrade prebiotics) that preferentially modulate the abundance of resident microbial species. Since mechanisms of pathogenicity are often complex and context-specific (21), it is preferred if these species are native to the microbiome of the host, as unwanted side-effects due to the introduction of foreign bacterial species might be avoided. Within this work, we will refer to prebiotics that specifically modulate the abundance of selected bacterial strains within the microbiome of a specific host as "precision prebiotics."

To design and evaluate such precision prebiotics, it is essential to start with a tractable model system such as the well-established model organism *Caenorhabditis elegans*. The native microbial community of the invertebrate *C. elegans* was first described in 2016 (22), yet the nematode has already been used for over 20 years as a model for studying host-microbial interactions (23). To similarly promote *C. elegans*-microbiome research, a synthetic 12-member *C. elegans*-specific microbiome resource, *CeMbio*, was recently established (24). Part of this consortium is the Gram-negative *Pseudomonas lurida* MYb11 that is known to possess antifungal (22) and antimicrobial properties, thereby providing

protection against the pathogen *Bacillus thuringiensis* (Bt) (21, 25). Besides its protective effects, MYb11 is known to colonize and persist in the *C. elegans* intestine (22) and thus increase the nematode's fitness during infection (25). *Ochrobactrum vermis* MYb71 is another member of the *CeMbio* community (24) and affects different host life-history traits (24, 26, 27), persistently colonizes the nematode (22), and has been described as competitive against other bacteria in terms of space and nutrients (27). Furthermore, the bacterial strain MYb11 was only able to grow on sucrose when co-cultured with MYb71 but not when cultured alone, suggesting that the two bacterial strains interact metabolically (27). Hence, a microbial community consisting of one beneficial bacterium and another one that is competitive offers an excellent opportunity for developing targeted approaches for microbiome manipulation.

A key prerequisite for the development of precision prebiotics is a detailed understanding of the metabolic functions present within a microbial community and how specific metabolic interventions affect microbiome metabolism. One important approach to investigate metabolic functions in microbial communities is constraint-based metabolic modeling (28–30). Based on the genetic information either on the level of each bacterium or from the overall community and biochemical databases, these methods allow one to simulate the network of chemical reactions that represent the metabolic reactions of the microorganism(s) or the community (31). To this end, genome-scale metabolic models of the bacterial species within a microbiome have to be reconstructed from genomic data. These models summarize the metabolic functions present within an organism and can be reconstructed using automated reconstruction approaches (32, 33) or obtained from existing model repositories (34). Furthermore, accounting for the growth environment of a species (35), optimization approaches such as flux balance analysis (FBA) for individual species (28) or constraint-based community modeling approaches (36, 37) can be used to predict metabolic activities within individual species as well as interactions between species. These methods have already been used to identify metabolic alterations in the microbiome of Parkinson's disease patients (38) as well as patients with inflammatory bowel disease (39) and to study the interaction of *C. elegans* as well as its microbiota in the context of drug-microbiome-host interactions (30). For *C. elegans*, metabolic models of the *CeMbio* community including MYb11 and MYb71 (24) and other members of the microbiota are available (27).

In this work, we used metabolic networks of MYb11 and MYb71 also incorporating phenotypic information from *Biolog* growth assays in a computational screen for precision prebiotics that are able to specifically boost the abundance of the host-protective strain MYb11 over MYb71. We tested four of the predicted compounds, L-serine, L-threonine, D-mannitol, and γ-aminobutyric acid (GABA) *in vitro* and confirmed that they selectively boost the abundance of MYb11. Moving to the *C. elegans* host system, we moreover showed that L-serine is able to selectively boost the abundance of MYb11 when the *C. elegans* host was colonized with both bacterial species. Thus, we demonstrate that constraint-based microbial community modeling approaches could represent a key tool for the design of targeted microbiome intervention approaches.

## MATERIALS AND METHODS

### Computational methods

#### Identification of unique uptake compounds

We explored to what extent individual metabolites, which are only taken up by a single bacterial species and no other, could serve as potential supplements to target individual species. To this end, we considered gut microbiome data from a Kiel-based human cohort comprising 1,280 participants for which 16S rRNA gene sequencing data is available, which we had utilized previously (30). As reported previously, the reads were mapped to the 16S sequences of the genomes of bacterial species present in the AGORA bacterial collection (34), the read counts of each sample were normalized to

a sum of 1 and removal of species below 0.1% abundance in each sample. We used *gapseq* version 1.1 (32) on the genomes of the identified species assuming the averaged dietary uptake of the human cohort (30) as input to the models to reconstruct genome-scale metabolic models. Subsequently, we identified for each bacterial species which metabolites can be taken up. To this end, for each exchange reaction of a metabolite, we set the respective reaction as an objective reaction. By convention, bacterial uptake of metabolites is modeled by exchange reactions where the forward direction corresponds to an excretion of a compound and the backward direction (i.e., negative flux) to its uptake. Therefore, we minimized the flux through each exchange reaction using FBA (40) implemented in the *R*-package *sybil* (41) which corresponds to maximizing the uptake of that metabolite. If the minimal flux through the exchange reaction was non-zero this metabolite could be taken up. Having repeated this procedure for each exchange reaction, we identified those compounds that can be taken up for each species. Then, we determined for each individual bacterial species in the gut microbiome of each participant whether it could take up a compound that could not be taken up by any other species in the bacterial community of this participant. Finally, we determined for each participant which fraction of the bacterial species contained in the gut microbiome had at least one unique uptake compound. Similarly, we reconstructed 78 bacterial models belonging to the *C. elegans* microbiota based on their published genomes (27) using *gapseq* version 1.2 (32). We then created random subsets of the models with increasing sizes (from 2 to 78, 50 iterations for each size) and determined the portion of strains with unique uptake compounds for each community. For the *CeMbio* community we used metabolic models reconstructed with *gapseq* 1.2 (32) from Zimmermann et al. (42). In general, the software *gapseq* (32) has an extensive published documentation regarding the reconstruction of genome-scale metabolic models, also available online (https://gapseq.readthedocs.io/en/latest/).

## Computational supplementation experiments

In our simulations, we utilized metabolic networks of *P. lurida* MYb11 and *O. vermis* MYb71 that were created from their genome sequences (27) using *gapseq* (32) version 1.1. Information from phenotypic microarrays about the organisms' metabolic capacities was integrated into the models using the "adapt" module of *gapseq*. Specifically, we incorporated data from the *Biolog* EcoPlate (assuming growth if OD590−OD750 >0.1) (24). During growth simulations, exchange fluxes of the models were constrained by assuming a computational nematode growth medium (NGM) (30) and aerobic conditions. These constraints can be found in Table S1. Additionally, the inflow of copper and iron cations was increased, so that they did not restrict the growth of the models. Following published practices (36) and due to the lack of the required kinetic information, we used the concentration values of the NGM medium input for the lower bounds of the models.

We used three distinct approaches for modeling individual microbes and microbial communities. To model the growth of each bacterium independently, we used FBA (40). We also utilized individual-based modeling based on *BacArena* (36) *R*-package and a community-FBA as described previously (39, 43, 44). The community-FBA approach is implemented in the *R*-package *MicrobiomeGS2* (www.github.com/Waschina/MicrobiomeGS2) and an overview has been described in detail in the publication by Pryor et al. (30). To measure the effects of supplements on growth of the individual species, we modified the published approach, so that we did not fix species composition but let it be determined as part of the optimization of individual species biomasses. In all types of simulations, we focused on the identification of nutritional supplements that boosted the growth of the target species MYb11.

For the individual supplementation experiments, the metabolic models of both organisms were constrained with the NGM computational diet, assuming that the molecular values of the dietary compounds (molecular quantities) can be directly incorporated into the models (as fluxes in the lower bounds of the models) without any

transformation. This is based on the assumption that the uptake rate of compounds is depicted by its maximum available amount (36). Supplementations for each metabolite were conducted by increasing the possible influx of the supplemented metabolite by 10 mmol/L. For testing the effects of supplements on single growth, the relative increase in growth rate compared to growth without supplements for each model was recorded. To prevent any numerical instabilities, all raw values were rounded to the sixth digit and relative changes bigger than 0.01 were taken into account.

For simulations using community FBA, the metabolic models of both organisms were merged into one microbial community, compartmentalized, model. Please note that we did not use a fixed community biomass composition as previously (30), since we employed community FBA to predict the resulting growth rates of individual species after maximizing the growth of the community. Thus, the community model was constrained with the NGM diet—extended with the individual supplements, as described in the previous paragraph. The growth rates of individual species were predicted by maximizing the sum of the growth rates of individual species with concomitant minimization of the sum of total fluxes as implemented in the *R*-package *sybil* (41) (with a coefficient of $-10^{-6}$ for the sum of fluxes in the objective function). Since the supplementations usually also increased the growth of the entire community, the predicted growth rates of each species were normalized with the respective growth rate of the community. Predicted growth rates following supplementation were contrasted against growth rates without supplementation to identify compounds whose effect was larger for the growth of MYb11 in comparison to MYb71. To exclude any numerical artifacts, rounding to the sixth digit and a cutoff of 0.01 in compound selection were also applied in this section.

For individual-based modeling of microbial communities, the metabolic models of both species were simulated in a virtual environment in *BacArena*, where they could randomly move and reproduce (36) on a 30 × 30 grid. As a growth medium, we used the NGM diet (in mM) which we diluted 1,000-fold, because otherwise bacterial models overgrew the entire simulation environment before nutrients were exhausted. For the simulations, we inoculated the growth environment with 20 individuals of each species and simulated bacterial growth for 12 iterations with 15 replicates for each supplementation using FBA. Replicates are needed since, in contrast to FBA, *BacArena* includes a stochastic component such as a random initial placement of individuals, random movement of individuals and random cellular replication events. A secondary objective to minimize the total flux while maximizing the growth rates of individuals was applied. To simulate the supplementation experiments, 0.01 mM of each compound of interest was added once in the beginning of the simulation. To identify significant changes in the growth profiles of individual species following each supplementation, we extracted the cell biomass information from the simulations using the command *plotGrowthCurve* and we calculated the relative change of the total biomass values (time step 13; rounding to the sixth digit) due the supplementations for each model. Then, the relative changes between the bacterial models were compared using Wilcoxon rank-sum tests (45). The resulting *P* values were corrected for multiple testing using false discovery rate control implemented in *R*. We selected those compounds whose effect on MYb11 was larger than on MYb71 based on the median of the respective relative changes for each model.

Based on these simulation techniques, we identified those compounds that specifically boosted MYb11 across the simulations. Subsequently, we subsetted this list to compounds for which we had positive confirmation of their uptake by MYb11 from *Biolog* experiments (24, 27). For the enrichment analysis of growth-supporting compounds of MYb11, we used the human metabolome database (HMDB) annotation for the available metabolite classes (46). Briefly, we tested whether the sets of candidate compounds were enriched in the metabolite classes defined in HMDB using Fisher's exact test. As the underlying set of metabolites, we used the metabolites contained in the metabolic model of MYb11 in the different metabolite classes.

### System information and software

Simulations and analyses were performed in the *R* environment (versions 4.0.0, 4.0.3, and 4.2.1). Model reconstructions were performed with the *gapseq* software (versions 1.1 and 1.2) (32) Optimizations were generally conducted using the software *sybil* (versions 2.1.5 and 2.2.0) (41). Specifically, multi-model supplementation was conducted using the software *MicrobiomeGS2* (version 0.1.5, www.github.com/Waschina/MicrobiomeGS2) and the in-silico co-culturing was simulated on *BacArena* (version 1.8.2, commit fdb02bf7) (36). Both incorporate *sybil* in their pipelines. The linear programming solver was *CPLEX* (versions 12.10.0 and 22.1.0), based on the *R* package *cplexAPI* (version 1.4.0). For data management, the software *dplyr* (versions 1.0.2, 1.0.4, and 1.0.9), *tidyr* (version 1.1.3), *tidyverse* (version 1.3.2) (47), and *data.table* (version 1.14.2) were used. For parallel computing, the software *foreach* (version 1.5.0 and 1.5.1) and *doMC* (version 12.10.0) were used. For plotting, the software *ggplot2* (versions 3.3.3 and 3.3.6) (48), *egg* (version 0.4.5), and *ggVennDiagram* (version 1.2.2.) were utilized.

## Experimental methods

### Bacterial strains

To determine bacterial growth in mono- and co-culture either in the presence or absence of supplementation, similarly to the computational section, the two naturally associated *C. elegans* microbiota isolates *P. lurida* MYb11 and *O. vermis* MYb71 were used. Fluorescently labeled MYb11::dTomato (21) and MYb71::GFP were used to distinguish the two bacterial species in a mixed community. For comparison, reverse labels were also tested, using MYb71::dTomato and MYb11::sfGFP, to ensure no impact of the fluorescent label (Fig. S2). Integrated fluorescent strains of MYb71 and the MYb11::sfGFP strain were constructed by transposon insertion at the *attTn7* site following published work (49). Briefly, two populations of *Escherichia coli* SM10 strains were prepared: (i) a donor strain containing either pTn7xKS-sfGFP (pTW415) or pTn7xKS-dTomato (pTW416) plasmids, and (ii) a helper strain containing the pTNS2 (pTW10) plasmid. Liquid cultures (5 mL LB) of donor, helper (both 37°C) and target strains (26°C) were grown with agitation overnight, subcultured, and collected at an OD 0.4–0.6. Tripartite mating mixtures consisted of equal parts (1:1:1), incubated overnight at 26°C, followed by selection on LB agar plates containing gentamicin (10 µg/mL) and IPTG (1 mM). Fluorescent colonies of MYb71 were verified by PCR and sequencing using two pairs of MYb71-specific primers to the glmS gene region: (i) oDB097 (5′-CGCTCTGATCGATGAGACC-3′) and oDB098 (5′-TTGCGCGGCTG RTCGAC-3′), and (ii) oDB097 and a transposon WP11 primer (5′-CACGCCCCTCTTTAATACG A-3′) (49). MYb11::sfGFP was similarly confirmed by PCR and sequencing, following our previous protocol (21).

Bacteria were thawed from frozen stocks for each experiment, streaked to tryptic soy agar (TSA) plates and grown for 16–42 h at 25°C. Bacterial cultures of MYb11 and MYb71 were grown in liquid NGM overnight at 28°C in a shaking incubator. Cultures were adjusted to $OD_{600} = 0.5$ with phosphate-buffered saline (PBS). We combined equal volumes of the adjusted cultures and used these as starting co-cultures for all experiments with the two strains, thus ensuring identical starting conditions across experimental treatments, as required for inference of an influence of the experimental manipulation on relative bacterial abundances.

### Supplementation in vitro

L-serine (1714.1), D-mannitol (8883.1), and L-threonine (T206.1) were purchased from Carl Roth (Karlsruhe, Germany), and GABA (A2129) from Merck (Darmstadt, Germany) and dissolved to 1 M or 0.5 M stocks in water, depending on the maximum soluble concentration. Liquid NGM was supplemented with a final concentration of 10 mM of the respective supplement. For supplementation assays *in vitro*, 5 µL of either mono- or co-culture of the bacteria was added to 95 µL of liquid NGM in a 96-well plate, in the presence or absence of supplement. Treatments were randomized with respect to plate

layout. Plates were incubated at 28°C for 24 h in an Epoch2 BioTek plate reader with double orbital shaking at 807 CPM. We assessed changes in bacterial abundances by quantifying the number of colony-forming units (CFUs) of a sample of the cultures plated out on agar, because this allows us to distinguish between the two strains, which is not possible using standard OD measurements. For this approach, 7 µL of different dilutions of each treatment bacterial suspension of respective mono- or co-cultures was plated onto TSA plates. The plates were stored at 25°C and CFUs scored after 24–48 h. Statistical calculations and plots were performed with *R* software (version 4.1.3). For all CFU assays *in vitro*, statistical comparisons were performed with Wilcoxon signed-rank tests (45).

## Supplementation in vivo

The *C. elegans* strain N2 was used for all *in vivo* experiments and was originally obtained from the Caenorhabditis Genetics Center (CGC). Worms were maintained on NGM plates inoculated with *E. coli* OP50 at 20°C following standard procedures (50). Assay plates were prepared by adding L-serine at a concentration of 0, 10, or 100 mM to NGM. Please note that serine can affect *C. elegans* foraging behavior (51). However and importantly for our proof-of-concept study, none of the numerous *C. elegans* foraging studies [including our own long-term experience (52–54)] indicated that this nematode is capable of a differential uptake of distinct bacterial strains from a mixed bacterial lawn. Therefore, any possible change in serine-induced nematode behavior is unlikely to cause any changes in the relative abundance of bacterial strains in the *C. elegans* gut—as long as identical starting cultures are used across experimental treatments, which we ensured with our experimental protocol. Overnight cultures of MYb11 and MYb71 were adjusted to $OD_{600}$ = 2.0 in PBS. Thirty to forty synchronized first instar larvae (L1) were grown on supplemented NGM plates inoculated with 125 µL of respective mono- or co-culture of MYb11 and MYb71. After 72 h, bacterial lawns were sampled by pressing the circular end of a 200-µL pipette tip into the lawn, avoiding the edges and any worms. The resulting disc was homogenized in PBS with zirconia beads in a Bead Ruptor 96 (Omni International) for 3 min at 30 Hz.

To assess the number of CFU, worms were washed off plates using M9-buffer + 0.025% Triton X-100, washed five times and paralyzed with 5 mM tetramisole to stop pumping of the worms as described in (24). Worms were surface-sterilized following (24), then washed two times in PBS to remove residual bleach. Worms were transferred to a new tube to determine the exact worm number (~20) and PBS was added to a final volume of 400 µL. Worms were allowed to settle and 100 µL supernatant was collected (supernatant control). Worms were homogenized in the remaining PBS using 1 mm zirconia beads in a Bead Ruptor 96 for 3 min at 30 Hz. Homogenized worms, lawns, and supernatants were serially diluted in PBS and plated on TSA plates. After up to 48 h at 25°C, the colonies were scored at the appropriate dilutions and the CFUs per worm were calculated. Each colonization experiment was conducted in three independent runs and equal biological replicates. Data were pooled across runs, and a mixed model controlling for runs was used to assess the effect of L-serine supplementation on bacterial colonization in *C. elegans*. For all L-serine *in vitro* and *in vivo* experiments, statistical comparisons were performed with generalized linear models, if not otherwise mentioned. We calculated the expected proportion of MYb11 in worms as the product of MYb11 proportions available on the lawns and the observed shift in bacterial proportions due to host filtering:

$$E(\text{Prop. MYb11 in supplemented worms}) = \left( \begin{array}{c} \text{Prop. MYb11 on} \\ \text{supplemented lawns} \end{array} \right) \times \frac{\text{Prop. MYb11 in worms}}{\text{Prop. MYb11 on lawns}}$$

Plots and statistical analysis were produced with *R* (version 4.1.3) and edited with Inkscape (version 1.1.2).

## RESULTS

### Overlapping metabolic niches between microbiome member species prevent the use of unique uptake compounds as precision prebiotics

The simplest approach to specifically boost the abundance of a particular bacterial species in a microbial community is to supply the community with a compound that is only taken up by the target species but no other species. Hence, these compounds correspond to unique metabolic niches for the respective bacterial species. We initially tested computationally whether such unique uptake compounds typically exist in bacterial communities, using bacterial communities from a human cohort as well as *C. elegans* (30). We evaluated for each bacterial species of a particular community whether it was able to take up a metabolite that no other species of the community could take up and determined the fraction of bacterial species in the community that had such unique uptake compounds (see Materials and Methods). For the human cohort, the bacterial communities comprised on average 26–34 bacterial species (25–75% quantiles). Within this community size, on average, 5–20% of bacterial species were inferred to possess unique uptake compounds that could potentially be used to target those species. For smaller communities, up to 50% of bacterial species and for larger communities, only 5–10% had unique uptake compounds (Fig. 1A). For *C. elegans*, we do not have matching data like for the human cohort with species abundance information for a large number of samples and genome sequences of the corresponding bacterial species for metabolic

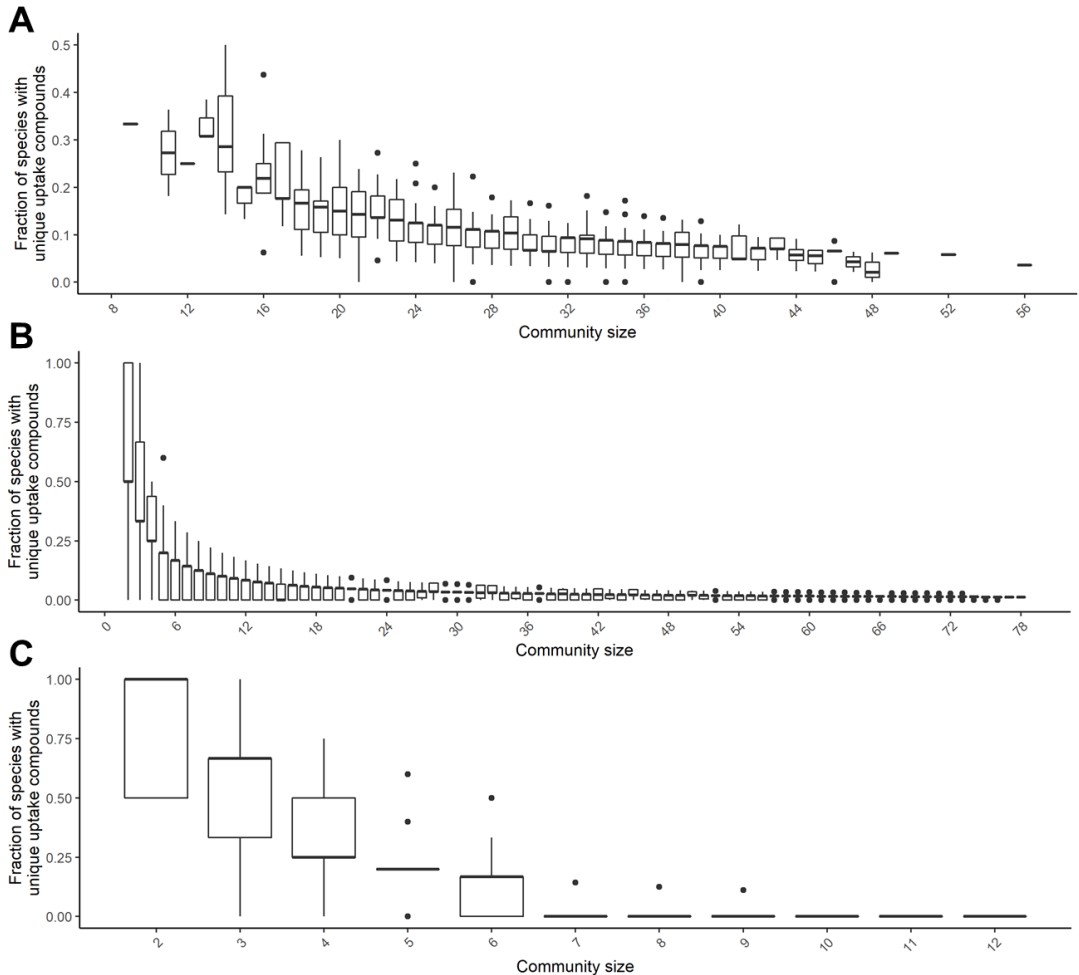

**FIG 1** Frequency of species with unique uptake compounds in microbiomes. Fraction of species with unique uptake compounds in (A) human gut bacterial communities and in (B and C) *C. elegans* bacterial communities that are subsets of previously reported *C. elegans* microbiome member species.

modeling. However, we can use strains that have been isolated from *C. elegans* and its environment as well as artificial minimal *C. elegans* communities to approximate the frequency of unique uptake compounds in the *C. elegans* microbiota. Thus, within a sample of 78 bacterial species isolated from *C. elegans* and its environment, we usually only identified 3–5% of species with unique uptake compounds in communities comprising 30 or more bacterial species, which is the typical size of bacterial communities observed in *C. elegans*-derived microbiome samples (55) (Fig. 1B). Similarly, in the artificial *CeMbio C. elegans* model microbiome community unique uptake compounds could usually only be detected for subsets of bacterial species up to a size of six member species (Fig. 1C).

Thus, in a typical microbiome like those analyzed above, we would expect to be able to identify unique uptake compounds only for a minority of bacterial species and compounds to target specific bacterial species are typically expected to be consumed by several species. Thus, for our identification of precision prebiotics for MYb11, we first excluded 20 compounds from our search that could specifically be taken up only by MYb11 in the search for a MYb-targeted prebiotic (20 compounds, see Table S2). Of note, MYb11 and MYb71 share the vast majority of all uptake compounds (85%), while only 10% and 5% of all uptake compounds are unique to MYb11 and MYb71, respectively (Fig. 2A).

## Computational identification of precision prebiotics targeting *P. lurida* MYb11

In order to derive precision prebiotics for *P. lurida* MYb11, we used its metabolic network as input for flux balance analysis and two distinct community modeling approaches apart from the basic approach of FBA. The first one is based on *BacArena* that incorporates an individual-based modeling approach and the other one is so-called community FBA implemented in *MicrobiomeGS2*. These two modeling approaches differ in their underlying assumptions. In *BacArena* the two bacteria are modeled by their individual metabolic networks and fluxes are predicted by maximizing the biomass fluxes of individual bacteria. In community FBA, in turn, the metabolic networks of the individual bacterial species are merged into a community model for which the sum of biomass fluxes of both bacterial species is maximized. Thus, when using *BacArena*, metabolic interactions between species typically arise from one species secreting an end-product that another species can metabolize. In community FBA, in turn, the optimization of total growth assumes that bacterial species coordinate their metabolic fluxes so that overall growth is maximized. Thus, since we do not know to which extent MYb11 and MYb71 mutually interact with each other, we cover both ends of the spectrum of potential interactions between species from completely individually optimized behavior in *BacArena* to an absolute coordination of fluxes to maximize overall community growth.

Using FBA, we identified 29 compounds that could increase the growth of MYb11 in single growth. Employing the community modeling approaches, we identified 81 compounds that increased the growth of MYb11 over MYb71 in community FBA and 17 compounds using *BacArena* (Fig. 2D). In order to determine whether specific compound classes were enriched among the MYb11-supporting compounds, we performed an enrichment based on HMDB annotation (46) across the three different simulation types. We found that growth-supporting compounds for MYb11 were enriched for fatty acids (Fisher's exact test *P* value = $3.16 \times 10^{-5}$).

Interestingly, the growth dynamics of the models when they were part of a community model (Fig. 2B) differ from the growth dynamics of the models when they were treated as individual organisms (Fig. 2C). For instance, MYb71's biomass flux was much higher in the community model, i.e., when coupled to the model MYb11. This is a typical observation when using community FBA when the biomass of the entire community is maximized. Especially for small communities, this often leads to one community member that most efficiently turns the growth medium into biomass to grow strongest with only little growth for other community members. On the other hand, in the *BacArena*

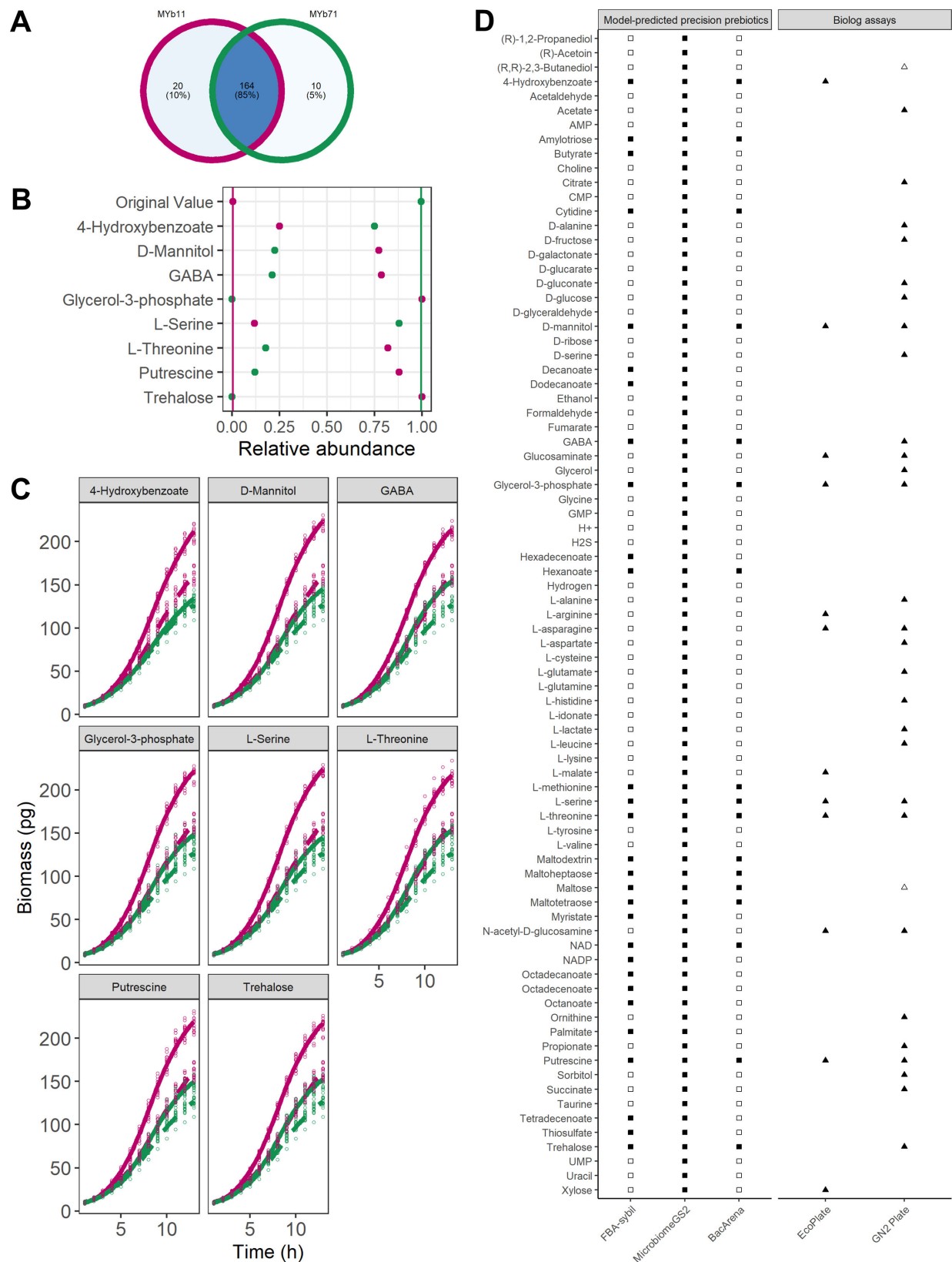

**FIG 2** Metabolic-modeling guided identification of precision prebiotics. (A) Overlap of all exchange reactions across the models MYb11 and MYb71. (B) Community FBA supplementation of *Pseudomonas lurida MYb11* (pink) and *Ochrobactrum vermis MYb71* (green) in *MicrobiomeGS2*. Predicted relative abundance of MYb11 and MYb71 under the supplementation with 4-hydroxybenzoate, D-mannitol, GABA, glycerol-3-phosphate, L-serine, L-threonine, (Continued on next page)

**FIG 2** (Continued)

putrescine, and trehalose shown in dots, lines represent predicted relative abundance without any supplementation. (C) *In-silico* supplementation of *Pseudomonas lurida MYb11* (pink) and *Ochrobactrum vermis MYb71* (green) in *BacArena*. Predicted growth curves of MYb11 and MYb71 either with 10 mM of 4-hydroxybenzoate, D-mannitol, GABA, glycerol-3-phosphate, L-serine, L-threonine, putrescine, or trehalose(solid line) or without supplementation (dashed line). (D) Comparison of predicted precision prebiotics for MYb11 in community with MYb71 with different computational approaches (first three columns) and growth data on *Biolog* EcoPlates and *Biolog* GN2 plates (24, 27) (last two columns). Full squares represent cases in which a metabolite was predicted as precision prebiotic for MYb11, empty squares if modeling predicted no increased growth of MYb11 over MYb71. For the *Biolog* data, filled triangles represent cases of catabolization of the respective compound and empty ones if no catabolization was observed. Missing circles indicate compounds not tested by Biolog plates.

simulations, we observed that most of those compounds boosted the growth of both species in single growth, however, the effect was larger for MYb11 than MYb71 (Fig. 2C). While *BacArena* also uses FBA to simulate bacterial growth, only a subset of FBA-predicted compounds that increase the growth of MYb11 over MYb71 was actually also identified by the *BacArena* approach (17 out of 29). The likely reason is that the FBA approach does not take into account potential interactions between species and that *BacArena* incorporates a stochastic component, for instance, for determining positioning as well as movement of bacteria and cellular replication events, such that small differences in growth rate between bacterial species might be filtered out by these stochastic events. Of the identified compounds, 17 compounds were shared across the three simulations. For eight of these, also information on uptake by MYb11 was available from *Biolog* plates which confirmed that MYb11 was able to consume them (24, 27) (Fig. 2D). Thus, combining modeling approaches and cross-referencing data from *Biolog* plates allowed us to considerably reduce our list of candidates to test experimentally to those giving the most consistent results. Based on the available literature information on the role of these compounds in longevity of the host (56, 57) or physiology (58), as nutritional compounds (24, 59), or compounds that are involved in host-microbial interactions (58, 60), we selected four candidates for further experimental testing as precision prebiotics to specifically increase the abundance of MYb11.

## *In vitro* validation of predicted precision prebiotics

As a proof-of-concept, we experimentally tested the ability of four of the predicted precision prebiotics compounds, L-serine, L-threonine, GABA, and D-mannitol, to specifically increase the growth of MYb11 in co-culture with MYb71. L-serine is a dietary amino acid that has been involved in nutrition-drug-host-microbial interactions (60) and can support the lifespan of the host (57), L-threonine, an essential amino acid (59), through its catabolite methylglyoxal (in low amounts) can also have a positive effect on host's lifespan (56). GABA, apart from being a bacterial product, is protective for the host's neurons (58). Finally, D-mannitol is a commonly used sugar alcohol (24), therefore its usage as a nutritional compound would make its usage as a supplement a safe option for the host.

We supplemented MYb11 and MYb71 in mono- as well as co-culture and measured bacterial growth *in vitro* in liquid NGM after 5 h (Fig. S1) and 24 h. Independently of the added supplement or the mono- or co-culturing, MYb11 and MYb71 showed increased growth in mono- and co-cultures when supplemented with 10 mM L-serine, L-threonine, GABA, as well as D-mannitol after 24 h (Fig. 3A; Tables S6 to S9). Additionally, to quantify bacterial growth, we used fluorescently labeled strains. CFU of mono- and co-culture were plated and counted after 24 h growth in liquid culture in the presence (10 mM) or absence (0 mM) of the four compounds L-serine (Fig. 3B; Table S10), L-threonine (Fig. 3C; Table S12), GABA (Fig. 3D; Table S13), and D-mannitol (Fig. 3E; Table S14). MYb11 and MYb71 showed increased colony numbers in mono-culture when supplemented with L-serine and L-threonine compared to no supplementation (Fig. 3B and C, Wilcoxon signed-rank test, $P = 0.001$, $P = 0.006$, $P = 0.007$, $P = 0.032$, respectively). GABA supplementation led to an increase in the growth of MYb11 in mono-culture, but not MYb71 (Fig. 3D, Wilcoxon signed-rank test, $P < 0.001$, $P = 0.922$, respectively), while D-mannitol had no effect on the growth of MYb11 nor MYb71 in mono-culture (Fig. 3E, Wilcoxon

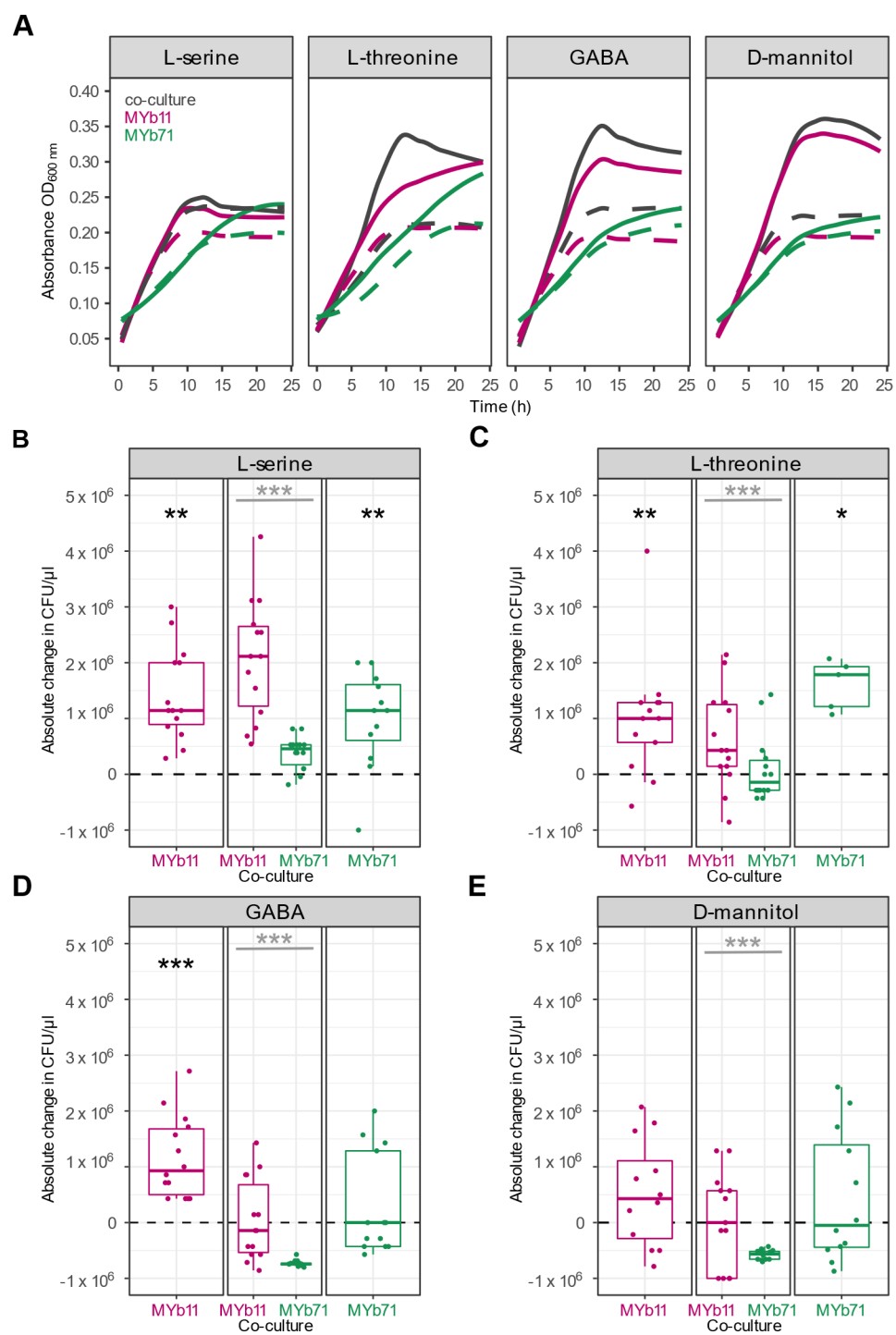

**FIG 3** *In vitro* growth of *Pseudomonas lurida* MYb11 and *Ochrobactrum vermis* MYb71 in mono- and co-culture with four different supplements for 24 h. (A) Growth curves of co-cultures (gray) or mono-cultures of MYb11 (pink) and MYb71 (green) in liquid NGM for 24 h either with 10 mM of L-serine, L-threonine, GABA, or D-mannitol (solid line) or without supplementation (dashed line). (B, C, D, E) Colony-forming units (CFUs/µL) in mono-cultures of MYb11 (left) and MYb71 (right) or in co-culture (middle) after 24 h of 10 mM of the respective supplement. Shown are boxplots with the median as a thick horizontal line, the interquartile range as box, the outer quartiles as whiskers, and each replicate depicted by a dot. The CFU count data of every replicate was normalized by subtracting the median of the CFU count data from the corresponding non-supplemented control treatments (control and experimental data provided in the Tables S10, S12 to S14). Black asterisks indicate statistical comparisons between supplemented and non-supplemented median, gray asterisks indicate statistical comparisons between supplemented medians of MYb11 and MYb71. $n = 5$–14.

signed-rank test, $P = 0.259$, $P = 0.661$, respectively). Thus, while serine as well as threonine boosted growth of MYb11 in co-culture versus unsupplemented single growth, and mannitol as well as GABA did not change growth in co-culture versus unsupplemented single growth, all four compounds specifically promoted MYb11 growth in co-culture over MYb71 *in vitro* and thus confirmed the computational predictions (Fig. 3B through E, Wilcoxon signed-rank test, $P = 6.87E{-}06$, $P < 0.001$, $P = 7.15E{-}06$, $P = 7.07E{-}06$, respectively) (see also: Table S11)

## Concentration-dependent effect of L-serine supplementation *in vitro*

The statistical analyses suggested that L-serine is the most promising precision prebiotic for further analysis among the compounds validated *in vitro*. We repeated the *in vitro* supplementation experiment across a concentration gradient to determine the optimal dose for enrichment of MYb11. Overall, the MYb11 growth as well as MYb71 growth in monoculture at 10 and 100 mM over time were increased following the increase in L-serine concentrations (Fig. 4A; Table S15). When quantifying the growth with

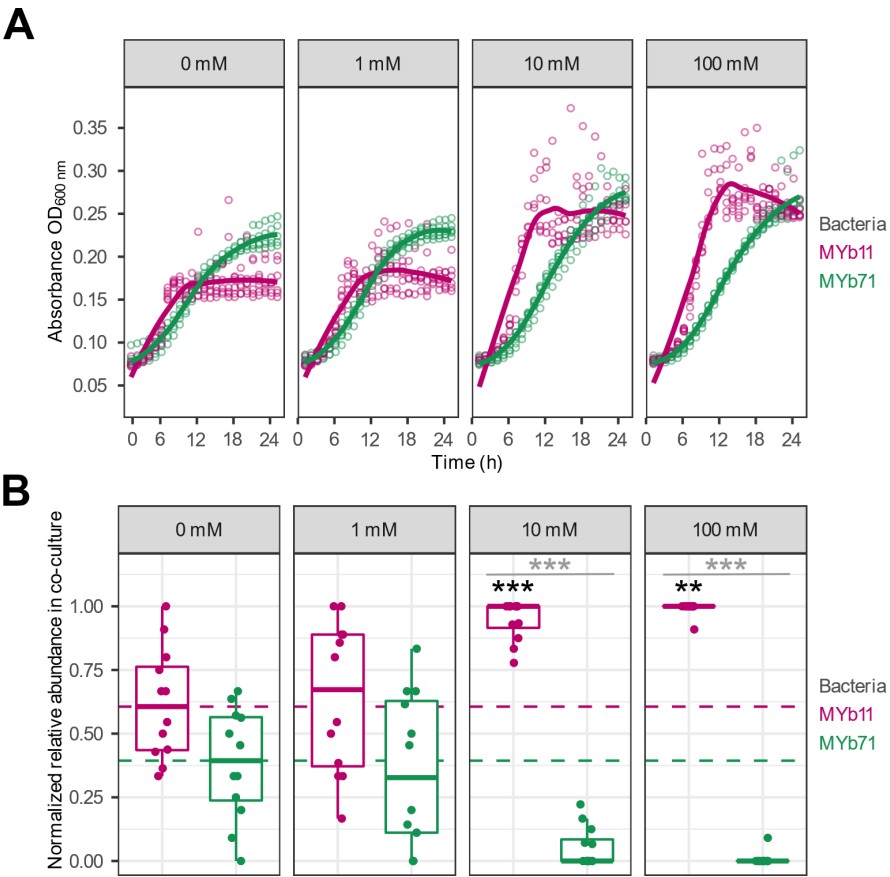

FIG 4 Concentration-dependent effect of L-serine supplementation on bacterial growth *in vitro*. (A) Bacterial growth was measured based on optical density ($OD_{600}$) of each species in mono-culture in liquid nematode growth medium (NGM) with the respective L-serine supplementation of 0, 1, 10, or 100 mM. Points represent individual replicates, while lines indicate the mean, $n = 12$. (B) Relative abundance of MYb11 (pink) and MYb71 (green) in co-cultures grown for 24 h with the respective supplement concentration in liquid NGM, based on colony counts of fluorescently labeled bacteria. Shown are boxplots with the median as a thick horizontal line, the interquartile range as box, the outer quartiles as whiskers, and each replicate depicted by a dot. Bacterial abundances of every replicate were normalized by subtracting the median of the corresponding non-supplemented control (0 mM) for each bacterial strain; the respective median values are indicated by the dashed lines. Statistical differences were determined by GLM and are indicated by asterisks (*** $P < 0.001$). Black asterisks indicate statistical comparisons between supplemented and non-supplemented median, gray asterisks indicate statistical comparisons between supplemented medians of MYb11 and MYb71. $n = 9$–$12$.

supplementation at the same concentration gradient (Fig. 4B; Table S16), then starting with 1 mM did not significantly impact the abundance of MYb11 proportions (GLM, $P$ = 0.778), nor MYb71 (GLM, $P$ = 0.69). However, supplementation with 10 mM led to an enrichment of MYb11 (GLM, $P$ < 0.001), thereby indicating a significant difference in proportions in co-culture between MYb11 and MYb71 (GLM, $P$ = 5.77E−07). Essentially identical results were obtained at 100 mM, including an MYb11 enrichment (GLM, $P$ = 0.004), and a significant difference between MYb11 and MYb71 proportions (GLM, $P$ = 7.93E−05). (See also: Table S17)

## Serine supplementation alters bacterial colonization of *C. elegans*

Finally, we tested whether L-serine supplementation to solid growth medium would alter bacterial proportions in the host *C. elegans*. Synchronized populations of L1 larvae were pipetted onto bacterial lawns of MYb11 and MYb71 on solid NGM plates supplemented with either 0, 10, or 100 mM L-serine. After 72 h, adult worms were collected to quantify bacterial proportions. Supplementation at 10 mM L-serine did not change the proportion of MYb11 in co-culture in worms compared to non-supplemented media (GLM, $P$ = 0.459, Fig. 5A left panel). However, L-serine supplementation at 100 mM increased the proportion of MYb11 in the bacterial community in worms compared to non-supplemented media (GLM, $P$ < 0.001, Fig. 5A, right panel). We next asked whether the effect of supplementation on bacterial proportions in the host simply reflected changes in the proportion of bacteria that were available to them on the lawn (i.e., their diet). The relative abundance of MYb11 on lawns did not change by 10 mM supplementation (GLM, $P$ = 0.61, Fig. 5A, left panel) compared to non-supplementation. However, the relative abundance of MYb11 was approximately 60% in non-supplemented media and increased to approximately 75% with 100 mM L-serine supplementation (GLM, $P$ < 0.002, Fig. 5B, right panel). The observed proportions of MYb11 in worms at 100 mM serine were significantly higher than expected based on proportions in the lawn and thus colonization probability compared to MYb71 (chi-squared test, $X^2$ = 10.244, $df$ = 1, $P$ = 0.0013; see Materials and Methods). This suggests that the provision of serine enriches MYb11 in the host independently from the enrichment observed on the lawn and that serine allows MYb11 to overcome its growth disadvantages in competition with MYb71 in the host (see also: Table S18 and S19).

## DISCUSSION

Although many factors are known to influence microbiome composition [e.g., dietary patterns, lifestyle, or medication (61)], the mechanisms causing these changes are often not fully understood and often a large number of bacterial species are affected. We propose that the most effective way to modulate the composition of the microbiome is by modulating or enriching the abundance of an already existing bacterial species with health-beneficial effects using precision prebiotics. A direct approach to identify such precision prebiotics would be to provide compounds that are uniquely taken up by the bacterial target species. However, through an analysis of *C. elegans* and human microbiome data, we were able to show that such unique uptake compounds typically only exist for few species in a microbial community and especially for larger communities there are no such compounds at all. Thus, in a typical microbiome, precision prebiotics have to be identified from compounds that are potentially taken up by several species. To identify such compounds, we used genome-scale metabolic networks of two bacterial members from the *C. elegans* microbiome, *O. vermis* MYb71, and the host-protective *P. lurida* MYb11, in conjunction with constraint-based modeling to identify compounds that specifically increased the growth of MYb11. Using HMDB annotation (46), we found that fatty acids were enriched as predicted growth-supporting compounds for MYb11 which is in agreement with previous observations of the preference of bacteria of the genus *Pseudomonas* for these compounds (62). Considering compounds that were identified across all three simulation approaches and cross-referencing with *Biolog* data revealed that 4-hydroxybenzoate, D-mannitol,

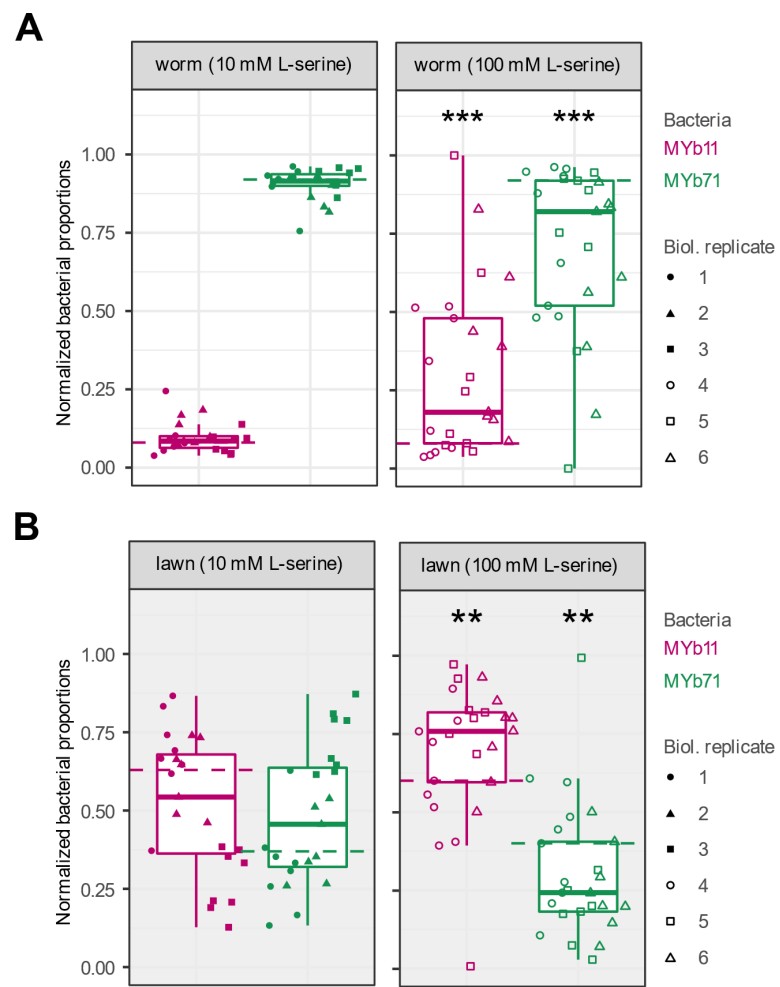

**FIG 5** L-serine supplementation alters bacterial abundance in *C. elegans*. (A) Adult worms were collected from solid NGM plates, which had been inoculated with co-culture and supplemented with 10 mM L-serine (left panel) or 100 mM (right panel) L-serine. Worms were washed and surface-sterilized to remove residual bacteria, then homogenized in a buffer. Bacterial proportions were quantified based on colony counts. (B) Bacterial lawns were sampled by cutting an agar disc of standardized size from NGM plates supplemented with 10 mM (left panel) or 100 mM L-serine (right panel). Discs were homogenized in buffer, and CFUs were quantified. (A and B) Shown are boxplots with the median as a thick horizontal line, the interquartile range as box, the outer quartiles as whiskers, and each biological replicate depicted by a symbol. For each replicate and each bacterium, bacterial abundance from experimental treatments was normalized by subtracting the median of the corresponding non-supplemented controls (0 mM), as indicated by the dashed lines. Statistical differences were determined by GLM and are indicated by asterisks (*** *P* < 0.001, \*\**P* < 0.01) and show comparisons between supplemented and non-supplemented median.

γ-aminobutyric acid (GABA), glycerol-3-phosphate, L-serine, trehalose, L-threonine, and putrescine are potential precision prebiotics for MYb11. We experimentally tested four of these predicted compounds, L-serine, L-threonine, D-mannitol, and GABA *in vitro* and confirmed their positive effect specifically on the growth of MYb11. Moving into the *C. elegans* host system, we were able to show that L-serine is also able to selectively boost the abundance of MYb11 *in vivo*, when the host was colonized with both bacterial species. Interestingly, while 10 mM of L-serine was sufficient for an enrichment of MYb11 in batch culture, 100 mM was necessary in the growth experiments with the host. This suggests differences in either bacterial uptake kinetics for L-serine or bacterial growth in general between batch growth and growth on agar plates. Moreover, while MYb11

was strongly enriched in worms to similar levels like MYb71 even though it had a very low abundance in the lawn, this enrichment was even stronger when supplying L-serine. Thus, our proof-of-principle work demonstrates that constraint-based microbial community modeling approaches are an effective tool for the future development of targeted microbiome therapies using precision prebiotics.

The effect of serine and threonine on the host and/or on its microbiome has been also documented in the literature. For instance, threonine is an essential amino acid for the host (59). Longevity effects on the host have been shown also after supplementation with threonine (63) and serine (57), while dietary serine reduced host fitness through acting on bacterial metabolism, when it was supplemented along with the chemotherapeutic 5-fluoro 2'deoxyuridine (60). Moreover, serine promotes the growth of *E. coli* LF82 in the mouse gut (64), which shows the potential of this compound to alter the abundance of multiple species. In accordance, we also observed the effects of serine on MYb71 growth, which was, however, stronger for MYb11 if they were grown in community.

One important challenge to specifically target the microbiome with nutritional supplements is to prevent host uptake or metabolization of the supplements (65). Different approaches to achieve restricted bioavailability, such as the supplementation of non-digestible carbohydrates (66) or coating of supplements with compounds that dissolve during passage through the digestive tract have been developed (67–69). Moreover, as indicated by our analysis of concentration-dependent effects of supplementation in the case of serine, it might be necessary to control the specific concentrations of a compound that are being released and their stability over time in the gut. From the bacterial side, *Biolog* assays have shown that MYb11 has the capacity to catabolize serine (Fig. 2D). However, the specific concentrations used on the *Biolog* plates and the mechanisms of transport that are possibly dependent on those concentrations are not known. From the host side, to maintain relatively constant concentrations in the host environment or to deviate from its normal concentrations, advanced formulations such as nanotechnology (70), delayed release, and other drug delivery technologies (71, 72) could be employed.

Although our combined computational-experimental approach can serve as a blueprint for future studies aimed at the identification of precision prebiotics, several factors should still be taken into consideration. As we showed, it is nearly impossible to identify a compound that is uniquely taken up by only one species within a community. This point becomes apparent, if we consider the numerous different species that reside in various hosts. For instance, the synthetic bacterial community of *C. elegans*, *CeMbio* (24) has 12 members, while the human-related bacterial community AGORA has more than 800 species (34). Thus, an important next step is to scale up our approach to more complex communities. Also along these lines, we did use automatically reconstructed bacterial metabolic networks from our gapseq approach rather than manually curated networks which resembles the typical situation in the human gut microbiota where a detailed manual reconstruction of the hundreds to thousands of already detected and sequenced bacterial species is practically impossible. Furthermore, since the physiology of worms cannot be directly compared to the physiology of humans, the efficacy and effectiveness of the approach [see more on the topic (73)] need to be studied in more detail. At this point, the complexity of the regulatory affairs in various jurisdictions (e.g., the definition of nutritional supplementation vs medication, safety concerns) should also not be underestimated (74). Additionally, while the modeling approach we used has quite a number of underlying assumptions, it allowed us to considerably narrow down the list of potential candidate metabolites for experimental testing. Finally, we did not consider potential side effects of the supplemented metabolites on the worm host and did not optimize the mode of delivery, for instance, to release the supplement specifically in the worm gut, which are important considerations for future studies. Taking everything into consideration, we showed that the *in silico* design of targeted

supplementation strategies to modulate the microbiome is not only feasible but also promising for the development of future microbiome-based treatment.

## ACKNOWLEDGMENTS

The authors acknowledge support by the German Research Foundation within the collaborative research center "Origin and Function of Metaorganisms," CRC1182, sub-project A1 to C.K., K.D., and H.S.; the research unit miTarget (FOR5042) to C.K. and the excellence cluster "Precision medicine in chronic inflammation" (EXC2167) to C.K. and H.S. C.K. moreover acknowledges support by the German Ministry for Education and Research (E:Med iTREAT, support code 01ZX1902A). G.M. received support through a Young Investigator Award by the CRC1182. R.D. received support through a Short-Term Research Grant from the Deutscher Akademischer Austauschdienst (German Academic Exchange Service). The authors also acknowledge funding via NIH grant DP2DK116645, NASA grant 80NSSC22K0250, and JGI/DOE grant CSP-503338 (all to B.S.S.).

This research was supported in part through high-performance computing resources available at the Kiel University computing center. *C. elegans* N2 was originally provided by the Caenorhabditis Genetics Center (CGC; https://cgc.umn.edu/), which is funded by the NIH Office of Research Infrastructure Programs (P40 OD010440).

## AUTHOR AFFILIATIONS

[1]Research Group Medical Systems Biology, University Hospital Schleswig-Holstein Campus Kiel, Kiel University, Kiel, Schleswig-Holstein, Germany
[2]Research Group Evolutionary Ecology and Genetics, Zoological Institute, Kiel University, Kiel, Schleswig-Holstein, Germany
[3]Department of Integrative Biology, University of California, Berkeley, California, USA
[4]Max-Planck Institute for Evolutionary Biology, Ploen, Schleswig-Holstein, Germany
[5]Alkek Center for Metagenomics and Microbiome Research, Baylor College of Medicine, Houston, Texas, USA
[6]Institute of Clinical Molecular Biology, Kiel University, Kiel, Schleswig-Holstein, Germany
[7]Institute of Diabetes and Clinical Metabolic Research, University Hospital Schleswig-Holstein Campus Kiel, Kiel, Schleswig-Holstein, Germany
[8]Nutriinformatics, Institute for Human Nutrition and Food Science, Kiel University, Kiel, Schleswig-Holstein, Germany

## AUTHOR ORCIDs

Georgios Marinos http://orcid.org/0000-0002-6443-7696
Inga K. Hamerich http://orcid.org/0000-0002-3451-4574
Reena Debray http://orcid.org/0000-0001-8130-4871
Nancy Obeng http://orcid.org/0000-0001-9639-2130
Carola Petersen http://orcid.org/0000-0002-2164-6789
Jan Taubenheim http://orcid.org/0000-0001-7283-1768
Johannes Zimmermann http://orcid.org/0000-0002-5041-1954
Dana Blackburn http://orcid.org/0000-0001-5153-3366
Buck S. Samuel http://orcid.org/0000-0002-4347-3997
Katja Dierking http://orcid.org/0000-0002-5129-346X
Andre Franke http://orcid.org/0000-0003-1530-5811
Matthias Laudes http://orcid.org/0000-0002-7846-955X
Silvio Waschina http://orcid.org/0000-0002-6290-3593
Hinrich Schulenburg http://orcid.org/0000-0002-1413-913X
Christoph Kaleta http://orcid.org/0000-0001-8004-9514

## FUNDING

| Funder | Grant(s) | Author(s) |
|---|---|---|
| Deutsche Forschungsgemeinschaft (DFG) | CRC1182 | Andre Franke |
| | | Hinrich Schulenburg |
| | | Christoph Kaleta |
| Deutsche Forschungsgemeinschaft (DFG) | FOR5042 | Andre Franke |
| | | Christoph Kaleta |
| Deutsche Forschungsgemeinschaft (DFG) | EXC2167 | Andre Franke |
| | | Matthias Laudes |
| | | Hinrich Schulenburg |
| | | Christoph Kaleta |
| Bundesministerium für Bildung und Forschung (BMBF) | 01ZX2202A | Christoph Kaleta |

## AUTHOR CONTRIBUTIONS

Georgios Marinos, Conceptualization, Formal analysis, Investigation, Methodology, Software, Writing – original draft, Writing – review and editing, Data curation, Funding acquisition, Project administration, Visualization | Inga K. Hamerich, Formal analysis, Investigation, Methodology, Writing – original draft, Writing – review and editing, Data curation, Software, Visualization | Reena Debray, Formal analysis, Investigation, Writing – review and editing, Data curation, Funding acquisition, Methodology, Software, Visualization | Nancy Obeng, Formal analysis, Investigation, Writing – review and editing, Conceptualization, Methodology, Supervision | Carola Petersen, Investigation, Writing – review and editing, Methodology, Software | Jan Taubenheim, Writing – review and editing, Methodology, Software | Johannes Zimmermann, Methodology, Writing – review and editing, Software | Dana Blackburn, Resources, Writing – review and editing, Methodology | Buck S. Samuel, Resources, Writing – review and editing, Funding acquisition, Methodology | Katja Dierking, Resources, Writing – review and editing, Funding acquisition, Methodology | Andre Franke, Resources, Writing – review and editing | Matthias Laudes, Resources, Writing – review and editing | Silvio Waschina, Software, Writing – review and editing | Hinrich Schulenburg, Conceptualization, Funding acquisition, Supervision, Writing – review and editing, Methodology, Resources | Christoph Kaleta, Conceptualization, Funding acquisition, Project administration, Supervision, Writing – original draft, Writing – review and editing, Formal analysis, Investigation, Methodology, Resources, Software

## DATA AVAILABILITY

The R scripts and the raw data to conduct the simulations are deposited in Github (version/commit: e15eb1e). Metabolic models used in the prediction of unique uptake compounds have been deposited in FigShare. The growth rate changes of the supplementation experiments with FBA sybil and MicrobiomeGS2 can be found in Supplementary Tables S3 and S4. The bacterial biomass data of the BacArena simulations can be found in Supplementary Table S5.

## ETHICS APPROVAL

The human cohort data used in this study originate from the publication of Pryor et al. (30), whose access was provided by the *PopGen* biobank (Schleswig-Holstein, Germany) (75). The raw data are available from the biobank through a structured application procedure (https://www.uksh.de/p2n/Information+for+Researchers.html).

## ADDITIONAL FILES

The following material is available online.

### Supplemental Material

**Supplementary figures (Spectrum01144-23-s0001.docx).** Figures S1 and S2.
**Supplementary tables (Spectrum01144-23-s0002.xlsx).** Tables S1-S20.

### Open Peer Review

**PEER REVIEW HISTORY (review-history.pdf).** An accounting of the reviewer comments and feedback.

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
