## [Reviewer comments · Microbiology Spectrum]

Microbiology Spectrum

Metabolic model predictions enable targeted microbiome manipulations through precision prebiotics

Georgios Marinos, Inga Hamerich, Reena Debray, Nancy Obeng, Carola Petersen, Jan Taubenheim, Johannes Zimmermann, Dana Blackburn, Buck Samuel, Katja Dierking, Andre Franke, Matthias Laudes, Silvio Waschina, Hinrich Schulenburg, and Christoph Kaleta

Corresponding Author(s): Christoph Kaleta, Christian-Albrechts-Universitat zu Kiel

Review Timeline:

Submission Date:	March 17, 2023
Editorial Decision:	June 30, 2023
Revision Received:	September 4, 2023
Accepted:	December 13, 2023

Editor: Paul Jensen

Reviewer(s): Disclosure of reviewer identity is with reference to reviewer comments included in decision letter(s). The following individuals involved in review of your submission have agreed to reveal their identity: Hyun-Seob Song (Reviewer #1)

Transaction Report:

DOI: <https://doi.org/10.1128/spectrum.01144-23>

June 30, 2023

Prof. Christoph Kaleta
Christian-Albrechts-Universität zu Kiel
Medical Systems Biology
Kiel
Germany

Re: Spectrum01144-23 (Metabolic model predictions enable targeted microbiome manipulations through precision prebiotics)

Dear Prof. Kaleta:

Thank you for submitting your manuscript to Microbiology Spectrum. Your manuscript was reviewed by two experts who have made several suggestions for improving the work. If you can address these points, I would be glad to consider a revised manuscript for publication. Please let me know if you have any questions or would like to discuss your revision plan.

Link Not Available

Sincerely,

Paul Jensen

Journals Department
Reviewer comments:

Reviewer #1 (Comments for the Author):

This work presents a combined use of three different models/simulations tools (FBA, community FBA (cFBA) and individual-based modeling (IbM)) to predict potential prebiotics that modulate relative abundances of a specific population in a microbial community. As a case study, the authors considered a binary consortium composed of *Pseudomonas lurida* MYb11 and *Ochrobactrum vermis* MYb71, two species chosen from *Caenorhabditis elegans* microbiomes. The effects of four of the predicted prebiotics were validated using in vitro and in vivo experiments. This work exemplifies the effective utilization of predictive models to guide the exploration and selection of potent prebiotics. Despite a great amount of work, however, the reviewer found some issues that need to be clarified for this paper to be accepted for publication.

1. The authors identified 29, 81, and 17 compounds that may promote the growth of MYb11 from FBA, cFBA, and lbM. Did they provide the full list of those compounds? How much was each of these predictions overlapping with their final selection (4-hydroxybenzoate, D-mannitol, γ -aminobutyric acid (GABA), glycerol-3-phosphate, L-serine, trehalose, L-threonine, and putrescine)?
2. The authors showed the validation for the four compounds (L-serine, L-threonine, GABA, and D-mannitol) against in vitro and only one (L-serine) against in vivo data. How about other compounds? Are there any reasons that the experimental and modeling results for other compounds were not presented? At least model predictions need to be presented. If model predictions were not consistent with experimental data for those compounds not included in the paper, please report as observed.
3. The more details need to be provided regarding the construction and FBA simulations of 78 bacterial models (in *C. elegans* microbiota).
 - a. It would be helpful if the authors provide an independent section in Methods to describe step-by-step procedures of metabolic network reconstruction so that the readers could also build the models following the given instruction.
 - b. Regarding FBA simulations using the R-package sybil, it is not clear how to get the LP solution. It is written that they minimized the flux through each exchange reaction, which corresponds to maximizing its uptake, but this is difficult for me to understand. A typical FBA first maximizes biomass production, followed by flux minimization - is this also what the authors did?
4. Issues with model accuracy:
 - a. Figure 2 shows significant discrepancy between Biolog data vs. growth predictions from three models. This indicates that individual metabolic networks constructed in this work were not correctly gapfilled. Why did the authors use incomplete models and how the results would be changed if they were fully gapfilled?
 - b. Difficult to understand why are the growth predictions between FBA-sybil and BacArena not the exactly same if they used the same metabolic networks?
5. The authors need to justify why the computationally expensive lbM model (that can account for spatial heterogeneity) was necessary. It seems that dynamic FBA would suffice. Were there any specific issues related to spatial heterogeneity to address in this paper?
6. In contrast with lbM, FBA and cFBA are steady state models that cannot predict "rates" but only "relative" fluxes to given uptake fluxes. The terms growth "rates" need to be corrected as appropriate.
7. In simulating lbM using BacArena, how were the authors able to determine the kinetic uptake rates of compounds (to use as upper or lower limits of exchange reactions)?
8. There are many different approaches for cFBA. More details need to be provided about MicrobiomeGS2 - does this take relative species abundances as input or predict them as model output? If the latter is the case, does it use a SteadyCom-type approach? If not, how?
9. How can the authors justify the following assumptions?
 - a. Maximization of community biomass in cFBA: Why do we expect two organisms to cooperative for maximizing the community biomass instead of individual biomass?
 - b. The assumption that the uptake rate of compounds is depicted by its maximum available amount: The amounts of compounds and their uptake rates are not the same.
10. More details need to be provided for the following:
 - a. "For simulations using community FBA, the metabolic models of both organisms were merged into one microbial community model" - I believe individual models were merged as a compartmentalized model (rather than mixed-bag), but it needs to be mentioned.
 - b. "For the simulations (of lbM), we inoculated the growth environment with 20 individuals of each species and simulated bacterial growth for 12 iterations with 15 replicates for each supplementation using FBA." - for those who are not very familiar with lbM simulations, please briefly explain why running BacArena needs iterations and replicates.
11. It would be beneficial if the authors publicly share all metabolic networks constructed in this work.

Reviewer #2 (Comments for the Author):

The presented study uses metabolic models, a defined coculture of bacteria, and *C. elegans* as a host for the coculture. The stated goal is to develop precision prebiotics to engineer microbiome composition.

The manuscript needs major revisions before it is ready for publication.

The document defines precision prebiotics as being able to target the abundance of a specific microbe in a microbiome for a rational goal. The document then states that precision prebiotics is not possible because of niche overlap between microbes found in natural microbiomes. The document then assembles a coculture and states that shifting the abundance of one microbe in the coculture now qualifies as an example of precision prebiotics. These arguments are contradictory and are not an effective way to motivate a study. The document needs to reduce its claimed impact and simply state that it outlines how to modestly change the frequency of a single microbe in a coculture by adding 10 g/L of serine.

The first section of the results section analyzes human microbiome metagenome data. This is a distraction from the rest of the paper which examines a *C. elegans* coculture. Please remove the human microbiome section. The data could be better applied to a different study further detailing the classic power law relationship between a community property and its frequency.

The document claims to use metabolic modeling to guide the study. The presented data does not support the argument that the modeling was meaningful. As acknowledged by the authors, there are major problems with the common consortia modeling approaches used in this study. The document addresses this shortcoming by applying three different approaches. However if all three approaches have major weaknesses, it is not clear if the aggregate output will be an improvement. To predict growth rates of specific consortia members requires a number of major and often fatal assumptions. First, the annotation of the transporter gene needs to be accurate enough to denote its substrate specificity which is challenging. Second, there needs to be major assumptions about the transporter kinetic parameters (v_{max}). Thirdly, the predicted biomass per substrate yield needs to be accurate which requires major assumptions about the utilized P/O number, maintenance energy requirements, and substrate preference. Fourth, the microbe needs to express a phenotype like lab grown *E. coli* that 'maximizes growth rate', this phenotype is likely less common away from laboratory conditions and rich media. Finally, all these assumptions need to be combined and applied to multiple organisms. The presented data indicate the applied assumptions were not reasonable. The presented experimental data, which must have been the best set of data, required 100 mM (10 g/L!) of serine to shift the population abundances in the coculture. If 10 g/L serine is required for a shift in outcomes, the cells likely do not have serine-specific transporters, instead the elevated concentration of serine likely enabled nonspecific transport into the cells. Typical K_m values for amino acids are ~1 mM, the tested concentration was 100x more than this. Secondly, in the medium supplemented with 10 g/L serine, the absorbance A600 of the target microbe only increased by ~0.15 units. For a typical bacteria, that corresponds with an increase of <0.1 g/L of biomass when 10 g/L of serine was added to the medium. 100x more serine was added than observed change in culture density. I am supportive of metabolic modeling but it must be applied and reported in a responsible manner else the field will earn a poor reputation, hurting all.

The document states that 100 mM of serine (10 g/L) results in a change in the frequency of the target species in the *C. elegans* host (figure 5). The presented data only shows a very modest change in species frequency and the target microbe is still the minority species in the host. How does 10 g/L of serine influence the phenotype of *C. elegans*? *C. elegans* has been documented to change its foraging behavior based on the presence of amino acids. Is 10 g/L of serine inhibitory to *C. elegans*? Are the results a product of host inhibition? At 10g/L, amino acid catabolism could result in a major increase in culture pH. What is the pH of the solid medium after growth?

Figure 5 plots data for the 10 and 100 mM serine conditions. There was not change for the 10 mM serine condition. The document states a control condition was also run but the data is not presented. The control data are essential to appreciate the effects of the serine and to gauge the natural variability in experimental data.

Experimental methods section, pages 13 and 14, it is stated that bacterial cultures of each strain were diluted to an OD of 0.5, then mixed together in equal volumes to make the co-culture. However, results are often measured in terms of CFUs of each strain. If the bacterial strains are different sizes, setting both cultures to the same OD could result in different numbers of bacterial cells being added to the initial co-culture, which would influence community composition throughout the experiment. Data in Figure 3 implies there is a different relationship between OD and cfu for the two species. Please report the OD to cell count conversions for each bacterial strain in order to confirm that mixing the cultures in this way does in fact result in equal proportions of each strain at the start of the experiments

Figures 3, 4, and 5, it is stated that each replicate was normalized by subtracting the non-supplemented median of the respective bacteria. Please explain why this was done. Does this overemphasize changes and that it confuses the interpretation of relative abundance? If the goal of the graphs is only to analyze change, the axes should be labeled accordingly. But this may also eliminate issues from the OD inoculation method?

Minor comments:

Figure 2. Please indicate that EcoPlate etc are Biolog data.

Introduction, sentence three, should be "host" rather than "hosts"

Introduction, paragraph three, please expand on what is meant by "e.g., microbial pathways of prebiotics degradation"

Page 7, please elaborate or explain the sentence "Both bacteria are also able to interact metabolically with the nutrient environment determining their type of interaction"

Page 8, should be "inflammatory bowel disease" rather than "inflammatory bowels disease"

Supplemental info, LEGEND tab, "constrains" should be "constraints"

Supplemental info, tab S3, please include raw data in addition to relative change in biomass flux. Please also explicitly describe how the relative change was determined (percent, fraction, etc). This seems to be stated in the final paragraph of page 10 but not clearly linked to the numbers in the supplemental data

Methods, page 11. It is stated that the reactions for the metabolic models were merged, but the biomass terms were kept

separate, and the effects of supplemented compounds were determined from the differences in the biomass fluxes. Maximizing the flux through one biomass term at a time in this community model assumes that these organisms are working together in order to increase the growth of both of them. Is this a reasonable assumption, given that MYb71 has been shown to be competitive? Also discussed page 20 in results section, where "coordinate" implies mutualism

Results, figure 2B, please change shape of markers so that the graphic can be read in black and white

Results, page 21. It is stated that the growth of 8 compounds was supported by Biolog data. Were the other compounds not included in the Biolog plate? Or was growth not shown in the assay?

Discussion page 28, sentence 1, should be "changes are" rather than "changes is"

Staff Comments:

Preparing Revision Guidelines

Please return the manuscript within 60 days; if you cannot complete the modification within this time period, please contact me. If you do not wish to modify the manuscript and prefer to submit it to another journal, please notify me of your decision immediately so that the manuscript may be formally withdrawn from consideration by Microbiology Spectrum.

Re: Spectrum01144-23R1 (Metabolic model predictions enable targeted microbiome manipulations through precision prebiotics)

Dear Prof. Kaleta:

I am pleased to inform you that your manuscript has been accepted. Thank you for your patience during review.

I am forwarding it to the ASM production staff for publication. Your paper will first be checked to make sure all elements meet the technical requirements. ASM staff will contact you if anything needs to be revised before copyediting and production can begin. Otherwise, you will be notified when your proofs are ready to be viewed.

Sincerely,
Paul Jensen
Editor
Microbiology Spectrum